# A Nomogram Based on Consolidation Tumor Ratio Combined with Solid or Micropapillary Patterns for Postoperative Recurrence in Pathological Stage IA Lung Adenocarcinoma

**DOI:** 10.3390/diagnostics13142376

**Published:** 2023-07-14

**Authors:** Longfu Zhang, Jie Liu, Dawei Yang, Zheng Ni, Xinyuan Lu, Yalan Liu, Zilong Liu, Hao Wang, Mingxiang Feng, Yong Zhang

**Affiliations:** 1Department of Pulmonary and Critical Care Medicine, Shanghai Xuhui Central Hospital, Shanghai 200031, China; longformany2000@sohu.com; 2Department of Pulmonary and Critical Care Medicine, Zhongshan Hospital, Fudan University, Shanghai 200032, China; liu.jie2@zs-hospital.sh.cn (J.L.); yang.dawei@zs-hospital.sh.cn (D.Y.); liu.zilong@zs-hospital.sh.cn (Z.L.); 3Department of Pulmonary and Critical Care Medicine, Zhongshan Hospital (Xiamen), Fudan University, Xiamen 361015, China; 4Department of Pathology, Zhongshan Hospital, Fudan University, Shanghai 200032, China; ni.zheng@zs-hospital.sh.cn (Z.N.); liu.yalan@zs-hospital.sh.cn (Y.L.); 5Key Laboratory of Public Health Safety, School of Public Health, Ministry of Education, Fudan University, Shanghai 200032, China; luxinyuan2211@163.com; 6Department of Thoracic Surgery, Zhongshan Hospital, Fudan University, Shanghai 200032, China; wang.hao@zs-hospital.sh.cn

**Keywords:** lung adenocarcinoma, stage IA, pathological subtype, nomogram, prognosis, consolidation tumor ratio

## Abstract

Background: Patients with pathological stage IA lung adenocarcinoma (LUAD) are at risk of relapse. The value of the TNM staging system is limited in predicting recurrence. Our study aimed to develop a precise recurrence prediction model for stage IA LUAD. Materials and methods: Patients with pathological stage IA LUAD who received surgical treatment at Zhongshan Hospital Fudan University were retrospectively analyzed. Multivariate Cox proportional hazards regression models were used to create nomograms for recurrence-free survival (RFS). The predictive performance of the model was assessed using calibration plots and the concordance index (C-index). Results: The multivariate Cox regression analysis revealed that CTR (0.75 < CTR ≤ 1; HR = 9.882, 95% CI: 2.036–47.959, *p* = 0.004) and solid/micropapillary-predominance (SMPP; >5% and the most dominant) (HR = 4.743, 95% CI: 1.506–14.933, *p* = 0.008) were independent prognostic factors of RFS. These risk factors were used to construct a nomogram to predict postoperative recurrence in these patients. The C-index of the nomogram for predicting RFS was higher than that of the eighth T-stage system (0.873 for the nomogram and 0.643 for the eighth T stage). The nomogram also achieved good predictive performance for RFS with a well-fitted calibration curve. Conclusions: We developed and validated a nomogram based on CTR and SMP patterns for predicting postoperative recurrence in pathological stage IA LUAD. This model is simple to operate and has better predictive performance than the eighth T stage system, making it suitable for selecting further adjuvant treatment and follow-up.

## 1. Introduction

Lung cancer remains the leading cause of cancer death worldwide, with lung adenocarcinoma (LUAD) being the most predominant subtype [1,2]. Surgical resection is the foundation of treatment in stage IA LUAD. However, some patients still suffered from recurrence despite complete resection. Patients with pathological stage IA non-small cell lung cancer have an estimated 5-year recurrence rate of 10–30% [3,4,5]. Thus, identifying high-risk groups of postoperative recurrence of pathological stage IA LUAD and carrying out closer follow-up and even adjuvant treatment is critical to improving the prognosis of patients.

Recent studies indicated that the consolidation tumor ratio (CTR) of lesions in pre-operation computed tomography (CT) scans was associated with recurrence rate. Wang et al. reviewed stage IA invasive adenocarcinoma manifesting as pure ground-glass nodule (CTR = 0) and found that there was no case recurrence within five years after resection [6]. A recently reported single-arm study (JCOG0804) showed that the 5-year recurrence-free survival (RFS) rate was 99.7% for lung cancer with a maximum tumor diameter ≤ 2.0 cm and with a CTR ≤ 0.25 [7]. Another randomized trial (JCOG0802) indicated the 5-year RFS rate was decreased to approximately 88% with a maximum tumor diameter ≤ 2.0 cm and with a CTR > 0.5 [8]. Zhai et al. reviewed 484 patients with pathological stage IA LUAD and found that the 5-year RFS rate was 79.2% for patients with pure solid nodules (CTR = 1) [9].

Besides, the adenocarcinoma subtype was associated with a relapse rate. According to the classification adopted by the World Health Organization (WHO) in 2015, adenocarcinoma subtyping was based on predominant histologic patterns [10]. Patients with solid or micropapillary predominant subtypes have poor prognoses [11]. Several studies have reported that mixed histologic patterns are common in LUAD. The presence of solid/micropapillary (SMP) features with minimal components can also be associated with an unfavorable prognosis [12,13]. Huang et al. analyzed 595 patients with pathological stage IA LUAD and divided them into three groups based on the total proportion of solid and micropapillary components (TPSM), which included TPSM-L (TPSM < 10%), TPSM-M (10% ≤ TPSM < 40%), and TPSM-H (TPSM ≥ 40%). The study found that patients with TPSM-H had a significantly lower 5-year RFS rate than those with TPSM-M or TPSM-L (51.5% vs. 72.2% or 85.0%, *p* < 0.001) [14].

Earlier, a nomogram was used in some studies to predict recurrence in patients with stage IA non-small cell lung cancer (NSCLC) [14,15,16,17,18]. However, the independent prognostic factors obtained by these studies are still controversial, and the prediction efficiency of most nomograms was not satisfactory (C-index < 0.8). Different studies have different definitions of high-risk subtypes, such as the proportion of solid/micropapillary (SMP) components ≥ 5%, ≥10%, or predominant [9,14,17]. The cutoff values of CTR are also in conformity. Yip et al. claimed that part-solids with CTR ≥ 0.8 were associated with lower RFS [19]. In another research, Sun et al. reviewed 415 patients with pathological stage IA LUAD and found a significantly worse RFS in those with a CTR > 0.5 [20].

In this study, we aimed to optimize risk stratification by CTR and SMP patterns on prognosis for patients with stage IA LUAD after surgical resection. Meanwhile, we identified the independent prognostic factors of RFS from other clinicopathological and CT parameters and developed a prognostic nomogram to facilitate the management of high-risk patients.

## 2. Materials and Methods

### 2.1. Patients

In the present study, 565 patients with pathological IA LUAD who underwent surgical resection at the Zhongshan Hospital Fudan University between January 2016 and December 2017 were retrospectively analyzed. This study was approved by the institutional review board of Zhongshan Hospital Fudan University. The informed consent from patients was waived because this was a retrospective study.

The inclusion criteria were as follows: (1) patients who were diagnosed as pathological IA LUAD; (2) patients who had primary lung adenocarcinoma as the pathological type; and (3) those with negative surgical margins (R0). The exclusion criteria were as follows: (1) patients with noninvasive pathological types, including adenocarcinoma in situ (AIS) and minimally invasive adenocarcinoma (MIA); (2) patients who received any preoperative or adjuvant anticancer therapy; and (3) patients with a history of other malignant tumors or multiple primary lung cancer. Clinical information, including age, gender, smoking history, and resection type, was also collected.

### 2.2. Radiological Evaluation

Computed tomography characteristics were reviewed independently by two chest radiologists, who were blinded to clinicopathological information. In the lung window, the maximum diameters of the solid tumor size and total tumor size were measured, respectively. The CTR was defined as the ratio of the solid tumor size to the total tumor size. The radiologists also determined whether the patient had emphysema.

### 2.3. Pathological Evaluation and Gene Testing

All surgical specimens were formalin-fixed and stained by hematoxylin-eosin (HE). Two pathologists with sufficient pulmonary pathology experience independently evaluated HE-stained sections. According to the eighth TNM staging system, the T stage was determined by the extent of tumor invasion measured by the pathologists. Each histological subtype (lepidic, acinar, papillary, solid, and micropapillary) was recorded based on the new WHO classification. The histological patterns were identified in 5% increments. The predominant pattern was defined as the pattern with the greatest percentage. We divided the patients into three groups based on the total proportion of solid/micropapillary components. Patients without a solid/micropapillary pattern were classified as solid/micropapillary-negative (SMPN), those with a solid/micropapillary component >5% but no predominant pattern as solid/micropapillary-minor (SMPM), and those with a solid/micropapillary component >5% and a predominant subtype as solid/micropapillary-predominant (SMPP).

EGFR mutations were screened in surgical specimens using the Amplification Refractory Mutation System–Polymerase Chain Reaction (ARMS-PCR) technology. Patients were screened for ALK rearrangement using immunohistochemistry (IHC). The Ki-67 expression of tumor samples was routinely detected using the Ki-67 protein antibody. Based on the previous study and according to the percentage of Ki-67 positive cells, two categories were defined: low Ki-67 expression (<10%) and high Ki-67 expression (≥10%).

### 2.4. Postoperative Follow-Up

The regular postoperative follow-up schedule included a chest CT scan and examination of lung cancer tumor markers every 3-6 months for the first two years and then once a year. Brain contrast-enhanced magnetic resonance imaging (MRI) and emission computed tomography (ECT) bone scans were recommended if clinically necessary. Telephone follow-up was performed as a complement. The RFS was defined as the period between the date of surgery and the date of the first event recurrence.

### 2.5. Statistical Analysis

Statistical analysis was performed using SPSS software for Windows version 24.0 (SPSS, Inc., Chicago, IL, USA). The chi-square test or Fisher’s exact test was used to compare categorical variables among different groups. Kaplan–Meier analysis was used to construct survival curves, and the log-rank test was used to assess the differences in survival. Univariate and multivariate Cox proportional hazard models were performed to identify independent prognostic factors for RFS. The nomogram was created using the R software (version 4.1.0) by integrating the independent prognostic factors identified in the multivariate Cox analysis. The accuracy of the nomogram was assessed by discrimination and calibration evaluation. The reported p-values were two-sided, with a *p*-value less than 0.05 considered statistically significant.

## 3. Results

### 3.1. Patient Characteristics

This study included a total of 565 patients. There were 347 women (61.4%) and 218 men (38.6%), with ages ranging from 24 to 86 years (median, 59 years). The predominant pathological subtype of the patients included lepidic (*n* = 102; 18.1%), acinar (*n* = 342; 60.5%), papillary (*n* = 79; 14.0%), solid (*n* = 30; 5.3%), and micropapillary (*n* = 12; 2.1%). Spread through air spaces (STAS) was found in 80 patients (14.2%). The CT characteristics of the patients included CTR ≤ 0.5 (*n* = 316; 55.9%), 0.5 < CTR ≤ 0.75 (*n* = 66; 11.7%), and 0.75 < CTR ≤ 1 (*n* = 183; 32.4%). Table 1 depicts detailed clinicopathological and CT characteristics.

### 3.2. Survival Analysis

The median follow-up time was 42.6 months. Additionally, postoperative recurrence was found in 34 patients (6.0%) and was counted as an event in the analysis of RFS. In all patients, the 3-year and 5-year RFS rates were 96.6% and 94.0%, respectively. We found that the 5-year RFS rate in patients with 0.75 < CTR ≤ 1 was significantly lower than those with 0.5 < CTR ≤ 0.75 or CTR ≤ 0.5 (84.2% vs. 95.5% or 99.4%, *p* < 0.001; Figure 1A). Furthermore, we found that the 5-year RFS rate in patients with SMPP was significantly lower than that of those with SMPM or SMPN (67.4% vs. 89.7% or 98.0%, *p* < 0.001; Figure 1B).

### 3.3. Independent Prognostic Factors

As shown in Table 2, sex, smoke, pathological stage, SMP, CTR, Ki-67 expression, and STAS were statistically significant in univariate COX analysis of RFS. Emphysema and EGFR mutation with a p-value less than 0.1 in the univariate COX analysis were further included in the multivariate Cox analysis where SMPP (HR = 4.969, 95% CI: 1.585–15.582, *p* = 0.006), and 0.75 < CTR ≤ 1 (HR = 11.541, 95% CI: 2.355–56.556, *p* = 0.003) were independent prognostic factors and negatively associated with RFS.

### 3.4. Development and Validation of the Nomogram

The nomogram was developed based on identified independent prognostic factors from the multivariate Cox analysis. The development of the nomogram for RFS illustrated CTR classification and SMP patterns (Figure 2), both of which contributed significantly to the prognosis, which showed good predictive value with an excellent C-index for RFS (0.873; 95% CI: 0.829–0.918). The C-index for RFS prediction by the eighth T stage alone was not satisfactory (0.643; 95% CI: 0.558–0.727). The C-index for RFS prediction by CTR (0.810; 95% CI: 0.765–0.855) and CTR combined with T stage (0.827; 95% CI: 0.775–0.879) were also inferior to the nomogram. The calibration plots revealed good consistency between the nomogram-predicted values and the observed outcomes, demonstrating that there was stability in predicting the survival of LUAD patients by the nomogram (3-year RFS: Figure 3A, 5-year RFS: Figure 3B). The predictive performances of different prognostic factors, including the prognostic nomogram, SMP patterns, CTR classification, and T stage, were compared using the ROC curve (3-year RFS: Figure 4A, 5-year RFS: Figure 4B). The results showed that the nomogram performed (AUC: 0.878, 0.926 for 3- and 5-year RFS) better than the SMP patterns (AUC: 0.842, 0.801 for 3- and 5-year RFS), CTR classification (AUC: 0.793, 0.910 for 3- and 5-year RFS), and T stage (AUC: 0.637, 0.707 for 3- and 5-year RFS). Therefore, our findings demonstrated that compared with the individual prognostic indicators, the nomogram is an excellent model for predicting the postoperative recurrence of stage IA LUAD.

## 4. Discussion

Patients with stage IA LUAD are always considered as having a good prognosis after surgical resection, but nearly 10–30% suffer from recurrence and death. Since stage IA LUAD is a group with high heterogeneity, it is of great significance to predict the prognosis for individual patients. Although previous studies have identified clinicopathological and CT predictors in early-stage LUAD, the present study has several unique aspects. First, we optimized risk stratification of CTR and SMP proportion on prognosis for patients with stage IA LUAD. Second, a comprehensive panel of clinicopathological and CT parameters, including STAS, emphysema, and gene mutation profiles, was collected. Multivariate Cox analysis was performed to identify independent prognostic factors of RFS. In essence, we developed a nomogram to quantify the risk of postoperative recurrence. This model is simple to use and has good predictive performance.

Previous research found that early-stage LUAD with ground glass opacification (GGO) features has a favorable prognosis [21,22]. CTR has been considered to be associated with outcomes in LUAD characterized by GGO. In this context, Zhai et al. divided stage IA LUAD into a GGO group and a pure solid group, where multivariate Cox analysis showed that GGO components were an independent prognostic factor for RFS. However, that study did not set an appropriate CTR cutoff value to predict postoperative recurrence, and the C-index of the nomogram was only 0.667 [9]. Yip et al. showed that for patients with a CTR < 0.8, the median RFS was greater than 97%, whereas, for the patients with a CTR ≥ 0.8, the median RFS was never greater than 86% [19]. Nevertheless, Yip et al. did not limit their study to patients with stage IA lung cancer. Sun et al. and Xi et al. reviewed stage IA LUAD manifesting as GGO and found that CTR > 0.5 was negatively associated with RFS [20,23]. The survival of stage IA LUAD patients with 0.75 < CTR < 1 was worse than those with 0.5 < CTR ≤ 0.75, but the difference was not statistically significant (*p* = 0.096). There was no relapse in the patients with a CTR ≤ 0.5 [24]. Therefore, we divided patients into three subgroups: CTR ≤ 0.5, 0.5 < CTR ≤ 0.75, and 0.75 < CTR ≤ 1. We found that the 5-year RFS rate in patients with 0.75 < CTR ≤ 1 was significantly lower than those with 0.5 < CTR ≤ 0.75 or CTR ≤ 0.5 (84.2% vs. 95.5% or 99.4%, *p* < 0.001; Figure 1A). Multivariate Cox analysis revealed that 0.75 < CTR ≤ 1 (HR = 11.541; 95% CI: 2.355–56.556, *p* = 0.003) was an independent prognostic factor for RFS, and our nomogram performed well with an excellent C-index for RFS (0.873; 95% CI: 0.829–0.918).

Several studies have reported that LUAD usually contains a mixture of histological subtypes, but defining a high-risk histological subtype remains controversial. In their multivariate Cox analysis, Kagimoto et al. analyzed 1059 patients with pathological stage 0 to III lung adenocarcinoma and found that the new grading system was not a predictive factor of RFS [24]. Although Zhai et al. found that a subtype with at least 5% solid/micropapillary presence was negatively associated with RFS, the C-index of the nomogram was 0.667 [9]. Huang et al. analyzed 595 patients with pathological stage IA LUAD and found that a subtype with at least 10% solid/micropapillary presence was a negative prognostic factor and had a C-index of 0.67 [14]. Qian et al. divided 1131 patients with stage IB LUAD into three groups: SMPN, SMPM, and SMPP. Multivariate Cox analysis showed that the SMPM and SMPP patterns were poor prognostic factors for RFS [25]. In our study, we also divided patients into three subgroups, including SMPN, SMPM, and SMPP. We found that the 5-year RFS rate in patients with SMPP was significantly lower than that of those with SMPM or SMPN (67.4% vs. 89.7% or 98.0%, *p* < 0.001; Figure 1B). Multivariate Cox analysis revealed that SMPP (HR = 4.969; 95% CI: 1.585–15.582, *p* = 0.006) was another independent prognostic factor for RFS, and our nomogram exhibited better predictive performance than previous studies.

According to the eighth TNM staging system, the T stage is the only variable used to subdivide stage IA LUAD. However, the present study showed that the predictive performance of the T stage alone for RFS is not satisfactory (C-index = 0.643). Here, the multivariate Cox analysis demonstrated that 0.75 < CTR ≤ 1 and SMPP were negative prognostic factors for patients with stage IA LUAD. The C-index increased to 0.873 when we developed a nomogram based on the two independent prognostic factors for predicting RFS. According to the ROC curve, the performance of the nomogram (AUC: 0.878, 0.926 for 3- and 5-year RFS) was vastly better than that of the T stage (AUC: 0.637, 0.707 for 3- and 5-year RFS). However, it is unclear why T stage and STAS were not independent prognostic factors for RFS. Similar to our study, previous studies found that the T stage was not an independent prognostic factor for RFS in stage IA LUAD [9,23,26]. Huang et al. indicated that it might be due to the difficulty distinguishing the tumor T stage [26]. Xi et al. performed logistic regression analyses, which confirmed that the T stage was not an independent risk factor for micropapillary or solid patterns, whereas CTR was the only risk factor for micropapillary or solid patterns [23]. Wang et al. analyzed 1387 patients with stage I NSCLC and found that STAS was associated with high-grade histological patterns. No STAS was recorded in tumors without a high-grade pattern component [27]. Thus, the influence of STAS on prognosis was based on pathological subtypes.

Previous studies have shown limited efficacy of adjuvant chemotherapy (ACT) in stage IA LUAD under surgical resection. Therefore, we need to screen the patients at high risk of recurrence for ACT. Based on the CTR classification and SMP patterns, we developed this nomogram model which had better predictive value than the previous study. We recommend close follow-up and potential postoperative adjuvant therapy for patients with a recurrence probability higher than 30%.

There are some limitations to our study. First, this study was a single-center retrospective study, which may have inevitable selection bias. Second, the observer variance of pathologists and radiologists may influence the judgment accuracy of CTR and SMP. Finally, this study only included single-center cases, and the prediction performance test was internal validation only, with no external validation. Further validation will require large-sample prospective multicenter studies in the future. We also hope that more relevant factors will be included in the future to improve the predictive performance of the nomogram model.

In conclusion, 0.75 < CTR ≤ 1 and SMPP were associated with negative prognosis of patients with stage IA LUAD. A nomogram based on the CTR classification and SMP patterns for RFS showed better predictive performance than the conventional T stage and CTR or SMP alone. This nomogram model can predict the risk of recurrence in individual patients and identify high-risk groups for close follow-up and adjuvant treatment.

## Figures and Tables

**Figure 1 diagnostics-13-02376-f001:**
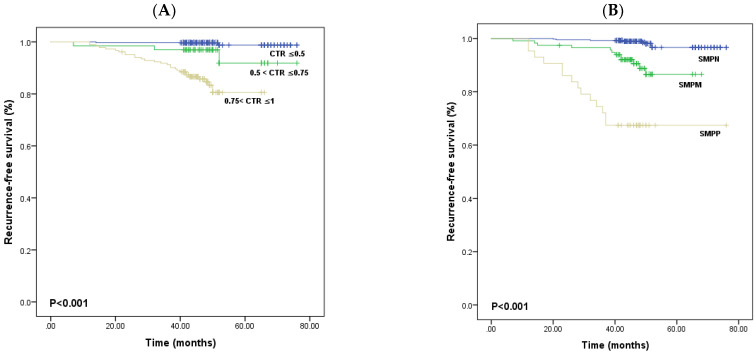
Kaplan–Meier survival curves for different CTR subgroups (**A**). Kaplan–Meier survival curves for different SMP subgroups (**B**). SMP, solid/micropapillary; CTR, consolidation tumor ratio.

**Figure 2 diagnostics-13-02376-f002:**
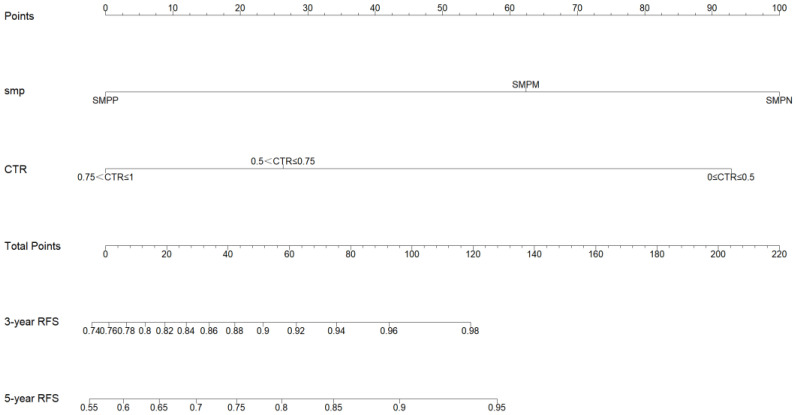
A nomogram for predicting RFS in stage IA patients with LUAD. RFS, recurrence-free survival; LUAD, lung adenocarcinoma. SMP, solid/micropapillary; CTR, consolidation tumor ratio.

**Figure 3 diagnostics-13-02376-f003:**
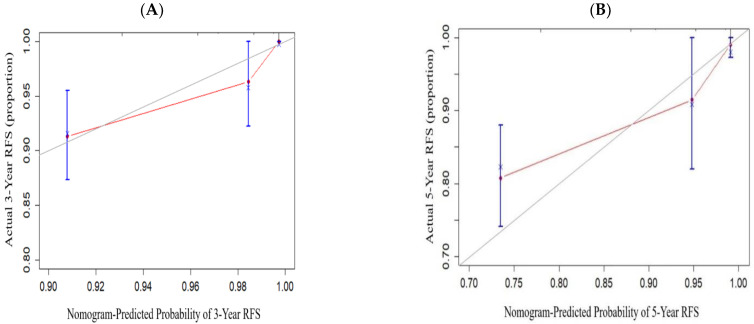
The calibration plots for predicting the 3-year and 5-year RFS in stage IA LUAD. (**A**) Calibration plots for 3-year RFS. (**B**) Calibration plots for 5-year RFS. RFS, recurrence-free survival; LUAD, lung adenocarcinoma.

**Figure 4 diagnostics-13-02376-f004:**
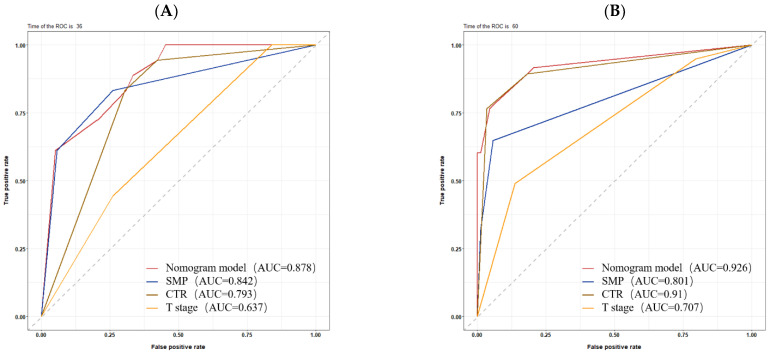
Receiver operating characteristic (ROC) curve analyses for predicting RFS in stage IA LUAD patients. ROC curves of the nomogram, SMP, CTR, and T stage for 3- and 5-year RFS (**A**,**B**). AUC, area under the curve; SMP, solid/micropapillary; CTR, consolidation tumor ratio.

**Table 1 diagnostics-13-02376-t001:** Characteristics of the included patients with stage IA lung adenocarcinoma.

Characteristics	Median or Case NO. (%)
Age (years)	59 ± 10.5
<65	374 (66.2)
≥65	191 (33.8)
Sex	
Male	218 (38.6)
Female	347 (61.4)
Smoke	
Ever	109 (19.3)
Never	456 (80.7)
Emphysema	
Absent	499 (88.3)
Present	66 (11.7)
CTR	
CTR ≤0.5	316 (55.9)
0.5< CTR ≤0.75	66 (11.7)
0.75< CTR ≤1	183 (32.4)
Pathological T stage	
T1a	88 (15.6)
T1b	326 (57.7)
T1c	151 (26.7)
Predominant subtype	
Lepidic	102 (18.1)
Acinar	342 (60.5)
Papillary	79 (14.0)
Solid	30 (5.3)
Micropapillary	12 (2.1)
SMP	
Negative	406 (71.9)
Minor	116 (20.5)
Predominant	43 (7.6)
Ki-67 expression	
<0.1	304 (53.8)
≥0.1	261 (46.2)
STAS	
Negative	485 (85.8)
Positive	80 (14.2)
EGFR mutation	
Negative	230 (40.7)
Positive	335 (59.3)
ALK rearrangement	
Negative	547 (96.8)
Positive	18 (3.2)
Resection type	
Lobectomy	314 (55.6)
Sublobar resection	251 (44.4)

CTR, consolidation tumor ratio; SMP, solid/micropapillary; STAS, spread through air spaces; EGFR, epidermal growth factor receptor; ALK, anaplastic lymphoma kinase.

**Table 2 diagnostics-13-02376-t002:** Univariate and multivariate analysis of RFS for pathological stage IA patients with LUAD.

Variables	Univariate Analysis		Multivariate Analysis	
	HR (95% CI)	*p*-Value	HR (95% CI)	*p*-Value
Age (years)				
<65	Reference			
≥65	0.85 (0.406–1.780)	0.667		
Sex				
Female	Reference		Reference	
Male	2.367 (1.195–4.687)	0.013	0.91 (0.334–2.476)	0.853
Smoke				
Never	Reference		Reference	
Ever	3.302 (1.664–6.552)	0.001	1.485 (0.548–4.022)	0.437
Emphysema				
Absent	Reference		Reference	
Present	2.206 (0.959–5.075)	0.063	0.723 (0.285–1.837)	0.496
Pathological T stage				
T1a	Reference		Reference	
T1b	2.049 (0.468–8.959)	0.341	0.705 (0.158–3.552)	0.717
T1c	5.306 (1.226–22.968)	0.026	0.97 (0.198–4.756)	0.97
SMP				
Negative	Reference		Reference	
Minor	6.442 (2.601–15.957)	<0.001	1.531 (0.514–4.561)	0.444
Predominant	25.182 (10.355–61.243)	<0.001	4.969 (1.585–15.582)	0.006
CTR				
CTR ≤0.5	Reference		Reference	
0.5< CTR ≤0.75	7.556 (1.262–45.230)	0.027	4.855 (0.763–30.908)	0.094
0.75< CTR ≤1	31.467 (7.457–132.778)	<0.001	11.541 (2.355–56.556)	0.003
KI-67 expression				
<0.1	Reference		Reference	
≥0.1	4.133 (1.870–9.136)	<0.001	1.063 (0.442–2.556)	0.891
STAS				
Negative	Reference		Reference	
Positive	5.192 (2.602–10.359)	<0.001	1.157 (0.561–2.387)	0.693
EGFR mutation				
Negative	Reference		Reference	
Positive	0.53 (0.270–1.039)	0.065	0.924 (0.427–1.999)	0.84
ALK rearrangement				
Negative	Reference			
Positive	0.047 (0–361.137)	0.504		
Resection type				
Lobectomy	Reference			
Sublobar resection	0.581 (0.283–1.192)	0.139		

CI, confidence interval; CTR, consolidation tumor ratio; SMP, solid/micropapillary; STAS, spread through air spaces; EGFR, epidermal growth factor receptor; ALK, anaplastic lymphoma kinase.

## Data Availability

The data presented in this study are available on request from the corresponding author. The data are not publicly available due to ethical reasons.

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
