# Peer review of "A Nomogram Based on Consolidation Tumor Ratio Combined with Solid or Micropapillary Patterns for Postoperative Recurrence in Pathological Stage IA Lung Adenocarcinoma"

_diagnostics, 2023, doi:10.3390/diagnostics13142376_

Round 1

Reviewer 1 Report

Dear Author:

I reviewed the manuscript entitled “A nomogram based on consolidation tumor ratio combined with solid or micropapillary patterns for postoperative recurrence in pathological stage IA lung adenocarcinoma.” The concept of nomogram described in this manuscript was interesting, but it has a few limitations that are addressed below.

Comments

1.      Both consolidation tumor ratio and micropapillary pattern had already been considered as  poor prognosis factors for lung adenocarcinoma based on a lot of previous papers. In this paper, a demonstration of nomogram based on these factors was interesting. However, these results were easily expected because these factors have been recognized as significant unfavorable variables in lung adenocarcinoma. Therefore, I didn’t think that there were so many impacts in this report. Based on these results, it is recommended that the authors add some comments which indicate some recommendations for procedural selection or adjuvant chemotherapy and so on in the discussion section.

2.      In line 20, is “our” “Our”?

3.      In this study, lobectomy and sublobar resection were equally included among resection types. Are there any selection criteria of resection types for lung cancer in your institution? I couldn’t understand the analysis of RFS included with these procedures together because the recent topic also has the procedural selection of stage IA lung cancer. I expected that the actual results of RFS would be different between lobectomy and sublobar resection.

Yours sincerely,

Author Response

We appreciated the thorough reviews provided by the journal and the positive comments from all reviewers. We have revised the manuscript in the light of their useful suggestions and comments. Below are our detailed responses to the comments and recommendations.

Response to Reviewer #1:

 I reviewed the manuscript entitled “A nomogram based on consolidation tumor ratio combined with solid or micropapillary patterns for postoperative recurrence in pathological stage IA lung adenocarcinoma.” The concept of nomogram described in this manuscript was interesting, but it has a few limitations that are addressed below.

  1. Both consolidation tumor ratio and micropapillary pattern had already been considered as poor prognosis factors for lung adenocarcinoma based on a lot of previous papers. In this paper, a demonstration of nomogram based on these factors was interesting. However, these results were easily expected because these factors have been recognized as significant unfavorable variables in lung adenocarcinoma. Therefore, I didn’t think that there were so many impacts in this report. Based on these results, it is recommended that the authors add some comments which indicate some recommendations for procedural selection or adjuvant chemotherapy and so on in the discussion section.

Many thanks for this advice and we added some comments which indicate recommendations for close follow-up and adjuvant chemotherapy in the discussion section.

Previous studies have shown limited efficacy of adjuvant chemotherapy (ACT) in stage IA LUAD under surgical resection. Therefore, we need to screen the patients at high risk of recurrence for ACT. Based on the CTR classification and SMP patterns, we developed this nomogram model which had better predictive value than previous study. We recommend close follow-up and potential postoperative adjuvant therapy for patients with a recurrence probability higher than 30%. (Page 9, line 291-296. Includes all modification flags)

  1. In line 20, is “our” “Our”?

 We really appreciate the careful reading,we have changed “our” to “Our”.

  1. In this study, lobectomy and sublobar resection were equally included among resection types. Are there any selection criteria of resection types for lung cancer in your institution? I couldn’t understand the analysis of RFS included with these procedures together because the recent topic also has the procedural selection of stage IA lung cancer. I expected that the actual results of RFS would be different between lobectomy and sublobar resection.

Thanks for your kind advice, our institution suggests that sublobar resection is an acceptable procedure for stage IA lung adenocarcinoma with peripheral lesions ≤2.0 cm and CTR ≤0.25. Because most of enrolled patients in our study were pure GGO or part GGO, this may be the reason why there was no difference in prognosis between lobectomy and sublobar resection.

Again, we really appreciated all the insightful comments. Thank you for taking the time and energy to help us improve the paper.

Reviewer 2 Report

This work represents a simple model that can be utilized for selecting further adjuvant treatments and follow-up. Few questions and justifications can further improve the study for an audience of interest. My comments are following. 

1.      Is the sample size of this single-center retrospective study sufficient to draw robust conclusions and develop a reliable nomogram for predicting recurrence-free survival (RFS) in stage IA lung adenocarcinoma? This point is not clear, please justify.

2.      Were there any potential sources of selection bias in the patient population, considering the retrospective nature of the study? How did the authors address or acknowledge this limitation?

3.      How did the authors ensure consistency and accuracy in the determination of solid tumor size (STS), computed tomography ratio (CTR), and the presence of micropapillary and solid patterns (SMPP) among different pathologists and radiologists involved in the study? Figure 1 needs more discussion related to the data shown.

4.      Has the developed nomogram been externally validated using independent datasets or multicenter studies to assess its generalizability and applicability in different patient populations?

5.      How do the predictive performance metrics, such as the concordance index (C-index) and area under the receiver operating characteristic curve (AUC), compare between the nomogram based on CTR and SMPP and the conventional factors like T stage alone or in combination with CTR?

6.      Were there any subgroups within the patient population that showed different predictive capabilities of the nomogram? For example, did the performance of the nomogram differ based on age, gender, or other clinical characteristics?

7.      Did the study consider the potential confounding effects of adjuvant treatments received by the patients on recurrence-free survival (RFS)? If so, how were these factors accounted for in the analysis?

8.      What are the potential implications of incorporating gene mutation profiles into the nomogram? Did the study identify specific gene mutations that significantly impacted RFS in stage IA lung adenocarcinoma?

9.      Considering the study's retrospective design, what future research directions do the authors suggest validating and further enhancing the predictive performance of the developed nomogram, such as large-sample prospective multicenter studies? This should be added in the discussion and in conclusion.

10.  How does this study contribute to the existing body of knowledge on prognostic factors and nomogram development for stage IA lung adenocarcinoma? What are the potential clinical applications and implications of the findings in terms of risk assessment, follow-up strategies, and adjuvant treatment decisions? There are many flaws in this work related to these questions that needs to improve. 

Author Response

We appreciated the thorough reviews provided by the journal and the positive comments from all reviewers. We have revised the manuscript in the light of their useful suggestions and comments. Below are our detailed responses to the comments and recommendations.

Response to Reviewer #2:

This work represents a simple model that can be utilized for selecting further adjuvant treatments and follow-up. Few questions and justifications can further improve the study for an audience of interest. My comments are following. 

  1. Is the sample size of this single-center retrospective study sufficient to draw robust conclusions and develop a reliable nomogram for predicting recurrence-free survival (RFS) in stage IA lung adenocarcinoma? This point is not clear, please justify.

Sample size calculation based on Cox regression. This procedure calculates power and sample size for testing the hypothesis that β1=0 versus the alternative that β1 = B. β1 is the change in log hazards for a one-unit change in X1 when the rest of the covariates are held constant. The procedure assumes that this hypothesis will be tested using the Wald (or score) statistic. Hsieh and Lavori (2000) gave a formula relating sample size, α, β, and B when X1 is normally distributed. The sample size formula is

In our study, CTR was used as the main research variable for sample size calculation, B=1.5, S=0.9, R2=0.375, P=0.06, alpha=0.05, and power=0.95. The sample size calculated using PASS software is 191, which is much smaller than the actual sample size included in this study.

  1. Were there any potential sources of selection bias in the patient population, considering the retrospective nature of the study? How did the authors address or acknowledge this limitation?

Our study was a single-center retrospective study, which may have inevitable selection bias. We acknowledge this limitation, further validation will require large-sample prospective multicenter studies in the future. (Page 9, line 296-301. Includes all modification flags)

  1. How did the authors ensure consistency and accuracy in the determination of solid tumor size (STS), computed tomography ratio (CTR), and the presence of micropapillary and solid patterns (SMPP) among different pathologists and radiologists involved in the study? Figure 1 needs more discussion related to the data shown.

STS, CTR and SMPP were reviewed independently by two pathologists and two radiologists, who were blinded to clinicopathological information. For patients with disagreement, a comprehensive discussion was conducted until a consensus was reached.

Thanks for your kind advice and we added more discussion related to the data shown in Figure 1.

We found that the 5-year RFS rate in patients with 0.75< CTR ≤1 was significantly lower than those with 0.5< CTR ≤0.75 or CTR ≤0.5 (84.2% vs. 95.5% or 99.4%, P < 0.001; Figure 1A). Multivariable Cox analysis revealed that 0.75< CTR ≤1 (HR = 11.541; 95% CI: 2.355–56.556, P = 0.003) was an independent prognostic factor for RFS, and our nomo-gram performed well with an excellent C-index for RFS (0.873; 95% CI: 0.829–0.918). (Page 8, line 251-255; Page 9, line 267-271. Includes all modification flags)

we found that the 5-year RFS rate in patients with SMPP was significantly lower than that of those with SMPM or SMPN (67.4% vs. 89.7% or 98.0%, P < 0.001; Figure 1B). Multivariable Cox analysis revealed that SMPP (HR = 4.969; 95% CI: 1.585–15.582, P = 0.006) was another independent prognostic factor for RFS, and our nomogram exhibited better predictive performance than previous studies. (Page 9, line 268-272. Includes all modification flags)

  1. Has the developed nomogram been externally validated using independent datasets or multicenter studies to assess its generalizability and applicability in different patient populations?

Many thanks for this advice,our nomogram haven't been externally validated using independent datasets or multicenter studies. We are planning a prospective multicenter study to validate its predictive performance.

5.How do the predictive performance metrics, such as the concordance index (C-index) and area under the receiver operating characteristic curve (AUC), compare between the nomogram based on CTR and SMPP and the conventional factors like T stage alone or in combination with CTR?

Many thanks for this advice, the predictive performances of different prognostic factors, including the prognostic nomogram, SMP patterns, CTR classification, and T stage, were compared using the ROC curve and C-index in the result and discussion.

The development of the nomogram for RFS illustrated CTR classification and SMP patterns (Figure 2), both of which contributed significantly to the prognosis, which showed good predictive value with an excellent C-index for RFS (0.873; 95% CI: 0.829–0.918). The C-index for RFS prediction by the 8th T stage alone was not satisfactory (0.643; 95% CI: 0.558–0.727). The C-index for RFS prediction by CTR (0.810; 95% CI: 0.765–0.855), and CTR combined with T stage (0.827; 95% CI: 0.775–0.879) were also be inferior to the nomogram. (Page 6, line 187-194. Includes all modification flags)

The predictive performances of different prognostic factors, including the prognostic nomogram, SMP patterns, CTR classification, and T stage, were compared using the ROC curve (3-year RFS: Figure 4A, 5-year RFS: Figure 4B). The results showed that the nomogram performed (AUC: 0.878, 0.926 for 3- and 5-year RFS) better than the SMP patterns (AUC: 0.842, 0.801 for 3- and 5-year RFS), CTR classification (AUC: 0.793, 0.910 for 3- and 5-year RFS), and T stage (AUC: 0.637, 0.707 for 3- and 5-year RFS). (Page 6-7, line 197-203. Includes all modification flags)

  1. Were there any subgroups within the patient population that showed different predictive capabilities of the nomogram? For example, did the performance of the nomogram differ based on age, gender, or other clinical characteristics?

Thank you for your question, the nomogram was established by integrating independent prognostic factors in the multivariate Cox analysis. Our study revealed that CTR classification and SMP patterns were the only two independent prognostic factors, there were no subgroups within the patients that showed different predictive performance of the nomogram.

  1. Did the study consider the potential confounding effects of adjuvant treatments received by the patients on recurrence-free survival (RFS)? If so, how were these factors accounted for in the analysis?

Thank you for your question, we have taken into account the effect of postoperative adjuvant therapy on RFS. Patients receiving any preoperative or adjuvant anticancer therapy is one of the exclusion criteria in our study. According to ASCO guidelines, there is almost no postoperative adjuvant therapy for stage IA lung adenocarcinoma in our hospital.

  1. What are the potential implications of incorporating gene mutation profiles into the nomogram? Did the study identify specific gene mutations that significantly impacted RFS in stage IA lung adenocarcinoma?

Thank you for your question, some previous studies have explored that gene mutation profiles influencing recurrence and metastasis in patients with stage IA LUAD. Gene mutation profiles were entered into the multivariate Cox analysis to indentify independent prognostic factors for RFS. However, the multivariate Cox analysis revealed that gene mutations were not independent prognostic factors for RFS.

  1. Considering the study's retrospective design, what future research directions do the authors suggest validating and further enhancing the predictive performance of the developed nomogram, such as large-sample prospective multicenter studies? This should be added in the discussion and in conclusion.

 Many thanks for this advice and we added the future research directions in the discussion and conclusion part of the revised manuscript.

Further validation will require large-sample prospective multicenter studies in the fu-ture. We also hope that more relevant factors will be included in the future to improve predictive performance of the nomogram model. (Page 9, line 301-304. Includes all modification flags )

  1. How does this study contribute to the existing body of knowledge on prognostic factors and nomogram development for stage IA lung adenocarcinoma? What are the potential clinical applications and implications of the findings in terms of risk assessment, follow-up strategies, and adjuvant treatment decisions? There are many flaws in this work related to these questions that needs to improve. 

Thanks for your kind advice and we have added contribution, clinical application, and limitations about our study in discussion of the revised manuscript.

Previous studies have shown limited efficacy of adjuvant chemotherapy (ACT) in stage IA LUAD under surgical resection. Therefore, we need to screen the patients at high risk of recurrence for ACT. Based on the CTR classification and SMP patterns, we developed this nomogram model which had better predictive value than previous study. We recommend close follow-up and potential postoperative adjuvant therapy for patients with a recurrence probability higher than 30%. (Page 9, line 291-296. Includes all modification flags)

Again, we really appreciated all the insightful comments. Thank you for taking the time and energy to help us improve the paper.

Round 2

Reviewer 1 Report

Dear Author:

I reviewed the revised manuscript entitled “A nomogram based on consolidation tumor ratio combined with solid or micropapillary patterns for postoperative recurrence in pathological stage IA lung adenocarcinoma.” 

Thank you for revising some of the points I asked for according to my comments. I agreed with your comments.

Yours sincerely,